# Experimental and Numerical Study of the Dynamic Response of XCC Pile–Raft Foundation under High-Speed Train Loads

Qiang Fu  and Jie Yuan *

Guangzhou Higher Education Mega Center, School of Civil Engineering, Guangzhou University, 230 Wai Huan Xi Road, Guangzhou 510006, China; cefuqiang@gzhu.edu.cn
* Correspondence: yuanj@gzhu.edu.cn

**Abstract:** A series of dynamic large-scale model tests and three-dimensional finite element analyses were conducted to investigate the dynamic response of track embankment and XCC pile-raft composite foundation in soft soil for a ballastless high-speed railway under moving train loads. The results indicate that the vibration velocity obtained from the FE numerical simulation agrees well with that from the model test in vibration waveform, amplitude, and frequency characteristics. The peak values corresponding to the passing frequency of train carriage geometry (lc = 25 m), bogie (lab = 7.5 m), and axle distance (lwb = 2.5 m) respectively reflect the characteristic frequencies of the train compartment, adjacent bogie, and wheel load passing through. The peak velocity significantly depends on the distance from the track center in the horizontal direction, of which the attenuation follows the exponential curve distribution. The vibration velocities decrease rapidly within embankment, show a vibration enhancement region from raft to the 1 m depth of foundation soil, then decreases gradually along the subsoil foundation, to a very low level at the bottom of the subsoil, which is much lower than that at the track slab and roadbed. The pile-raft composite foundation can reduce the vibration level effectively and improve the safety of trains running in soft soil areas.

**Keywords:** high-speed train loads; XCC (X-section cast-in-place concrete) pile–raft foundation; large-scale model test; soft soil; vibration propagation; attenuation

## 1. Introduction

The coupled interaction between railway tracks and substructures needs to be properly considered for the design of high-speed railways over soft clays [1–7], as it tends to result in geotechnical problems, such as a reduction of the bearing capacities of foundations and unexpected settlement. The working performance of the railway track, substructures, and underlying soils depends not only on the properties of individual components, but also on the coupled interaction between each other. The train-induced vibration of the track and ground is strongly affected by the relationship between the train speed and the corresponding propagating wave velocity of the supporting media. The speed at which large amplification of the dynamic response occurs is named the 'critical speed' [7]. At the critical speed, moving train loads induce strong vibration in the track structure, and increase the risk of train derailment and track structure damage. Madshus C, Kaynia AM [1] indicated that large dynamic amplifications appear in the dynamic response of the rail/embankment/ground system as the train speed approaches an apparently critical value. The critical speed is controlled by the minimum phase velocity of the first Rayleigh mode of the soil and embankment profile at the site.

The train's speed has an obvious effect on the dynamic responses of railway substructures that are constructed over soft clays [8–10]. Trains moving at high speeds generate large stress intensities at the soil layers, further causing permanent settlements. Embankments of soft soils supported by piles and high-strength geosynthetics have advantages for reducing the settlements of highways and railways. Stronger dynamic stresses induced

in soils underneath the track structure by faster trains will inevitably cause larger permanent settlements. Takemiya and Bian [8] proposed a substructure approach to study the dynamic interaction of track structures and layered soils under moving train loads. They indicated that the excessive response at high speeds should be controlled by applying the soil improvement technique at the track as demonstrated from the computer simulation by Takemiya [2], Takemiya and Kojima [4], and actually taken at the site as reported by Smekal and Berggren [3]. M.J.M.M. Steenbergen et al. [11] developed an analytical model to assess the design parameters of a slab track railway system from a dynamic viewpoint. They indicated that for high frequencies, an increase of the track stiffness is most effective, whereas for low frequencies, soil improvement is a better solution. Soft ground improvement using piles has increasingly been used as a rapid construction technique for high-speed railway embankments over soft soil areas. The bear capacity of piles and pile–soil interaction effect under different loading conditions have been studied by Achmus M. and K. Thieken [12] and Conte, E.; Pugliese, L et al. [13]. Extensive studies have been carried out to obtain insight into the mechanisms of the geosynthetic reinforced pile supported (GRPS) embankment by considering the pile–soil interactions [14–18]. Han GX et al. [16] studied the properties of soil arching under a dynamic load by using numerical methods and model tests. Tang Y et al. [19] developed a three-dimensional finite element method (FEM) to study the dynamic responses of GRPF under high-speed rail loads. Naveen Kumar Meena et al. [20] established a two-dimensional plane model to investigate the influence factor on soil arching for a pile-supported railway embankment subjected to train-induced loadings.

Analytical, numerical, and experimental methods have been conducted to investigate the mechanical behavior and dynamic interactions of piled raft foundations subjected to dynamic load [21–25]. Ramon Varghese et al. [23] presented a substructuring method-based soil structure interaction (SSI) analysis model to study the dynamic impedances of a piled raft (PR) in homogeneous and layered soil conditions. Sun G et al. [26] presented physical model tests on a ballastless track supported by a XCC pile–raft foundation, to investigate the vibration velocity on air-dried and saturated sand subsoils. The test results showed that the vibration velocity in a BTXPR foundation is closely related to the degree of saturation of the subsoil. The above scholars performed detailed studies on pile-supported embankment under dynamic loading and provided some valuable conclusions. However, theoretical, numerical, and experimental studies on the dynamic response character, load transfer mechanism, and pile–soil interactions of pile-supported foundations under dynamic loading are fewer than that of static loading, and very limited attention has been paid investigation of their vibration characters and behaviors under moving train loads. Vibration transmission is a complex process for railway track and pile su-ported embankment modeling because of the difficulty in representing the vibration source, the unknown damping coefficients along the propagation path, pile–soil interactions, and the lack of comprehensive information of the substructures.

Recently, X-section cast-in-place concrete (XCC), as a new type of noncircular cross-sectional shaped pile, was developed and is widely used in China for soft soil improvement and structural support [26–30]. The results show that piles with an X-shaped cross section can provide greater bearing capacity than traditional circular piles with the same section area. Lv et al. developed a series of analytical and numerical solutions for capturing the stress transfer mechanism and bearing capacity of XCC pile foundations under a vertical static load [27–29]. Sun G et al. [29] presented physical model tests on a ballastless track XCC pile–raft foundation at a scale of 1/5 under different applied cyclic load frequencies, to investigate the vibration velocity on air-dried and saturated sand subsoils. The test results showed that the vibration velocity in a BTXPR foundation is closely related to the degree of saturation of the subsoil. However, most studies conducted so far have investigated the behavior of the XCC pile–raft foundation under static loads, with very limited attention paid to the behavior of the foundation system under dynamic loads. The dynamic bearing capacity, pile–raft interaction effect, frequency response characteristics, and load transfer mechanism of a pile–raft foundation for high-speed railways under moving train loads are still not very clear.

In this paper, a series of large-scale model tests and three-dimensional finite element analyses for an XCC pile–raft composite foundation were conducted to investigate the dynamic responses of the XCC pile–raft composite foundation for a ballastless high-speed railway under moving train loads. In the FE numerical simulation, the equivalent moving M-shaped loading wave was used to simulate the moving train load by applying the load on the unit point in the numerical simulation, of which the time and frequency domain features are consistent with the loading waves in the model test. The model tests and FE simulation results are presented and compared regarding the variation of the dynamic velocity to analyze the characteristics of the dynamic response, transmission, and attenuation for the XCC pile–raft composite foundation.

## 2. Model Test of the XCC Pile–Raft Foundation for a Ballastless High-Speed Railway

### 2.1. Model Tests System

The XCC pile–raft foundation for a ballast-less railway was constructed in a reinforced concrete box with dimensions of 5 m × 4 m ×7 m (length × width × height), as shown in Figure 1. The track structure was composed of double rails, fasteners, a track slab, and a concrete base. Rails of CHN60 type (similar to UIC60-type rails) were fixed to the track slab with WJ7-type fasteners. The physical model was scaled to 1/5. The concrete base was constructed in situ with steel reinforcement. The substructure consists of embankment, concrete raft, gravel cushion, subsoil, XCC piles, and a supporting layer. The embankment was a layer of 0.54-m-thick AB granular below the concrete base. A concrete raft with the dimensions 5 m × 3.34 m × 0.12 m was constructed with steel reinforcements below the embankment. The gravel cushion was layered 0.06 m thick between the raft and the pile. The pile was 4.3 m (21.5 in full scale) in scaled size. The pile distance was 0.6 m. The underlying ground was composed of silty soil. The 1-m-thick supporting layer was composed of medium-density sand. The geometric parameters of the pile head are 2 R, 2a, and $\theta$, where 2R is the diameter of the surrounding circle, 2a is the length of the flat sides, and $\theta$ is the angle between tangents drawn from the center of two adjacent sides. In this study, the geometric parameters 2R, 2a, and $\theta$ were 0.157 m, 0.039 m, and 90°. The concrete strength grades of the XCC pile, raft, concrete base, and track slab are C25, C35, C40, and C55, respectively. The composite geo-membrane and asphalt material were used around the inner wall of the concrete box, to reduce the influence of dynamic wave reflection and friction force of the inner wall.

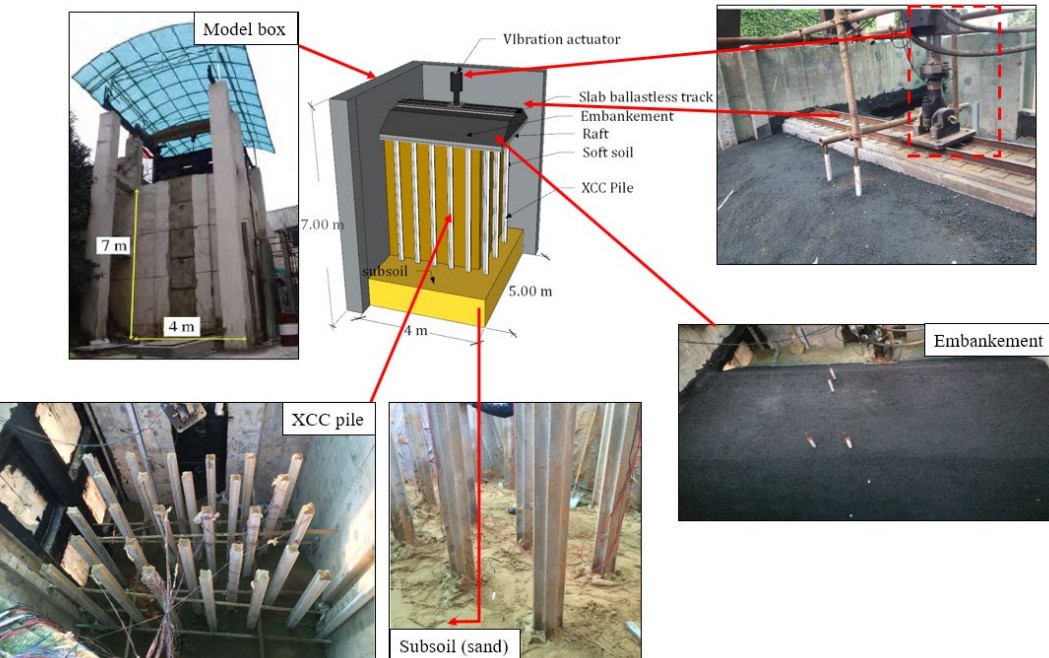

**Figure 1.** Physical model of ballastless track embankment and XCC pile–raft foundation.

The schematic arrangement of the velocity and soil pressure sensors used to record the vibration velocity and soil pressure is shown in Figure 2a,b. Velocity sensors (V1–V12) were placed in various locations along the horizontal and vertical directions.

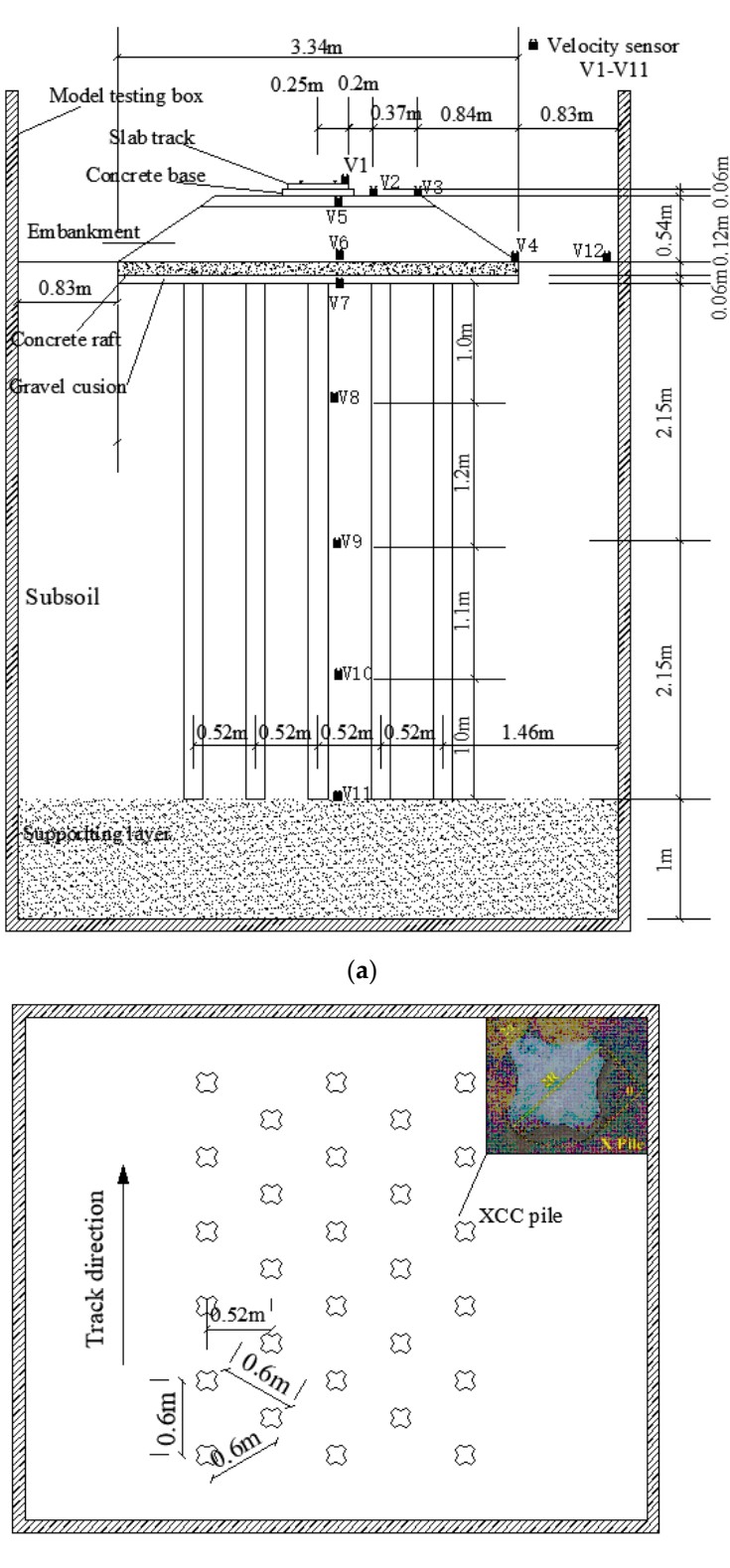

(a)

(b)

**Figure 2.** Geometry and instrument layout of the physical model test. (**a**) cross-sectional view, (**b**) plane view.

The grain-size distributions of gravel, granular soil, sand, and silty soil are plotted in Figure 3, and the physical properties and parameters of these geomaterials are summarized in Table 1.

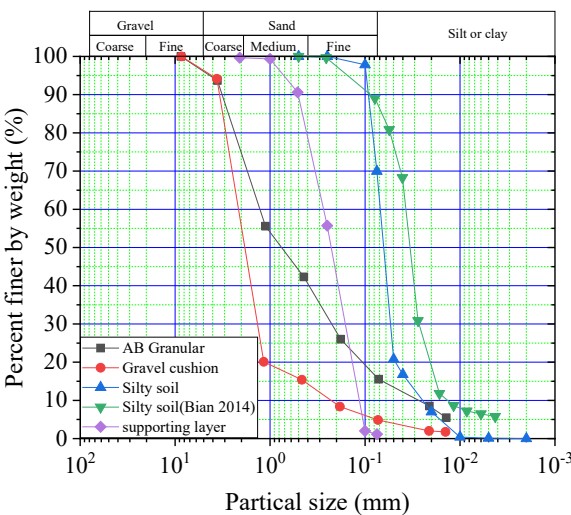

**Figure 3.** Grain-size distributions of geomaterials in ballastless railways.

**Table 1.** Physical properties and parameters of the geomaterials used.

| Filling Materials | $\gamma$ (kN/m³) | $w$ (%) | $Dr$ (%) | Liquid Limit (%) | Plastic Limit (%) | Plastic Index | $Cu$ | $Cc$ |
|---|---|---|---|---|---|---|---|---|
| Silty soil | 18.6 | 27.8 | — | 31.0 | 24.1 | 6.9 | 2.76 | 1.71 |
| AB Granular | 21.9 | 5.35 | 63 | — | — | — | 6 | 2.89 |
| Gravel | 23.51 | 7.07 | 92 | — | — | — | 19.23 | 1.34 |
| Supporting layer | 19.43 | 25.5 | 69.5 | — | — | — | 2.42 | 0.93 |

The embankment was compacted by a tamping machine to the designed densities to bear the static and dynamic loadings induced by the train. The compaction coefficient $K$, modulus of subgrade reaction $k_{30}$, and dynamic deformation modulus $E_{vd}$ were tested and are illustrated in Table 2. Although the physical model was reduced to the scale of 1/5, it still preserves the physical and mechanical properties of the natural materials. The scale factors of the dynamic model test were calculated by the Bockingham $\pi$ theorem and are summarized in Table 3.

**Table 2.** Tested results of the embankment.

| Parameters | $K$ | $K_{30}$/(MPa/m) | $E_{vd}$ (MPa/m) |
|---|---|---|---|
| Value | 0.97 | 243 | 98 |

**Table 3.** Scale factors of the dynamic model test.

| Geometry | *Load* | *Stress* | *Volume* | *Frequency* | Density | Velocity | Time | Length | Elastic Modulus |
|---|---|---|---|---|---|---|---|---|---|
| 1/5 | 1/25 | 1 | 1/125 | 5 | 1 | 1 | 1/5 | 1/5 | 1 |

### 2.2. Dynamic Loading System

A train's structure is composed of the carriage, bogie, train wheel, and other structures. The dynamic inertial effects are mainly associated with the train speed and train geometry, such as the dominant axle distance, bogie distance, and carriage length. The dynamic responses of substructures are caused by a series of trains' moving loads. Adjacent trains

have a cyclic superposition effect. The experiments simulated the cyclic passages of trains by applying loads on the railway with a certain shift in time. Every cycle of the applied loads corresponded to the passage of one carriage. The geometry configuration of the standard carriage of a China Railways high-speed train is shown in Figure 4a. The carriage length $l_c$ is 25 m, axle distance $l_{wb}$ is 2.5 m, and bogie distance $l_{ab}$ is 7.5 m.

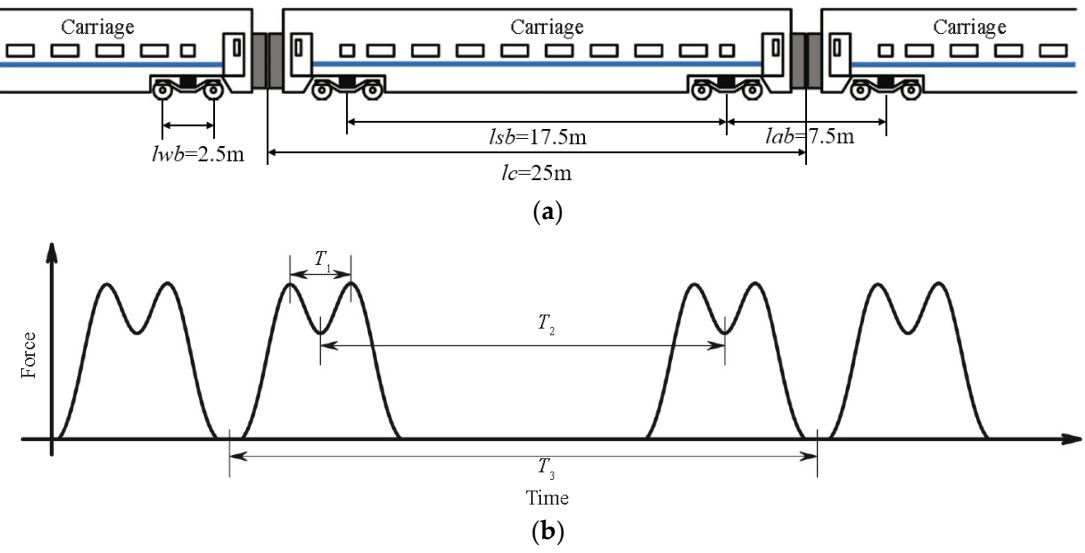

**Figure 4.** Geometry configuration of the high-speed trains and M-shaped wave. (**a**) Geometry configuration of the high-speed trains, (**b**) M-shaped wave.

Theory and empirical evidence from reported studies (Guangchao Sun et al. [26], Al Shaer et al. [31], Bian et al. [32], Zhi-Ping Z et al. [33]) show that the waveform of the dynamic load of a high-speed strain is similar to the shape of M, with each peak corresponding to the load exerted by an axle of the bogie. These two dominant frequencies of the dynamic response correspond to the length of a carriage (25 m) and the distance of two bogies in the adjacent carriages (7.5 m) (Bian et al. [32]). The experimental results from the full-scale model testing show good agreement with the field measurements, indicating that the applied equivalent loadings at the fastener positions generated by the proposed sequential loading system can replace the train's moving loads at the rail surface in a realistic high-speed railway.

The M-shaped dynamic loads generated by the electric hydraulic actuator were applied on the concrete slab track, to simulate the dynamic loading process of high-speed moving train loads. The cyclic load $F$ in the full-scale model can be described by the following three Fourier equation:

$$
\begin{aligned}
F(t) = a_0 + a_1 \cos(\omega t) + b_1 \sin(\omega t) + a_2 \cos(2\omega t) + b_2 \sin(2\omega t) \\
+ a_3 \cos(3\omega t) + b_3 \sin(3\omega t)
\end{aligned}
\tag{1}
$$

where $F(t)$ is the axle load. When the axle load is 157 kN, the train speed $v$ is 225 km/h and the simulated coefficients in the Fourier series are: $a_0 = 85.52$, $a_1 = -71.84$, $a_2 = -32.29$, $a_3 = 12.89$, $b_1 = 15.24$, $b_2 = 14.97$, $b_3 = -10.45$. The relationship between $\omega$ and different train speeds is shown in Table 4.

**Table 4.** Relationship between $\omega$ and different train speeds.

| Train Speed $V$ (km/h) | 45 | 90 | 135 | 180 | 225 | 270 |
|---|---|---|---|---|---|---|
| $\omega$ (rad/s) | 10.06 | 20.128 | 30.192 | 40.25 | 50.321 | 60.387 |

The time interval $\Delta t$ between two axles in the track's direction can be expressed as follows:

$$\Delta t = \frac{l_{wb}}{v} \tag{2}$$

where $l_{wb}$ is the axle distance and $v$ is the train speed.

One cycle loading time T between two carriages can be expressed as follows:

$$T = \frac{l_c}{v} \tag{3}$$

The M-shaped wave considering the entire carriage effect is shown in Figure 4b, where $T_1$ is the time that two wheel sets in a bogie traverse a location ($l_{wb}/v$), $T_2$ is the time that two bogies in a carriage traverse a location, and $T_3$ is the time that a carriage traverses a location.

Trains running at different speeds were simulated by adjusting the loading cycle time T and the time parameter $\Delta t$ (see Table 5).

**Table 5.** Characteristic parameter of the actuator loading time–history curve.

| Train Speed $v$ (km/h) | 45 | 90 | 135 | 180 | 225 | 270 |
|---|---|---|---|---|---|---|
| Peak time $T_1$ (s) | 0.2 | 0.1 | 0.0667 | 0.05 | 0.04 | 0.0333 |
| Cycle time $T_3$ (s) | 2 | 1 | 0.667 | 0.5 | 0.4 | 0.3333 |
| Peak frequency (Hz) | 5 | 10 | 15 | 20 | 25 | 30 |

After a series of experiments, simulation and normalization correction to the axle loads were inducted. The applied load on the actuator is specified as 5 kN for the scaled model with load scale factors of 1/25. The actuator loading time–history curve was generated for different train speeds, as shown in Figure 5.

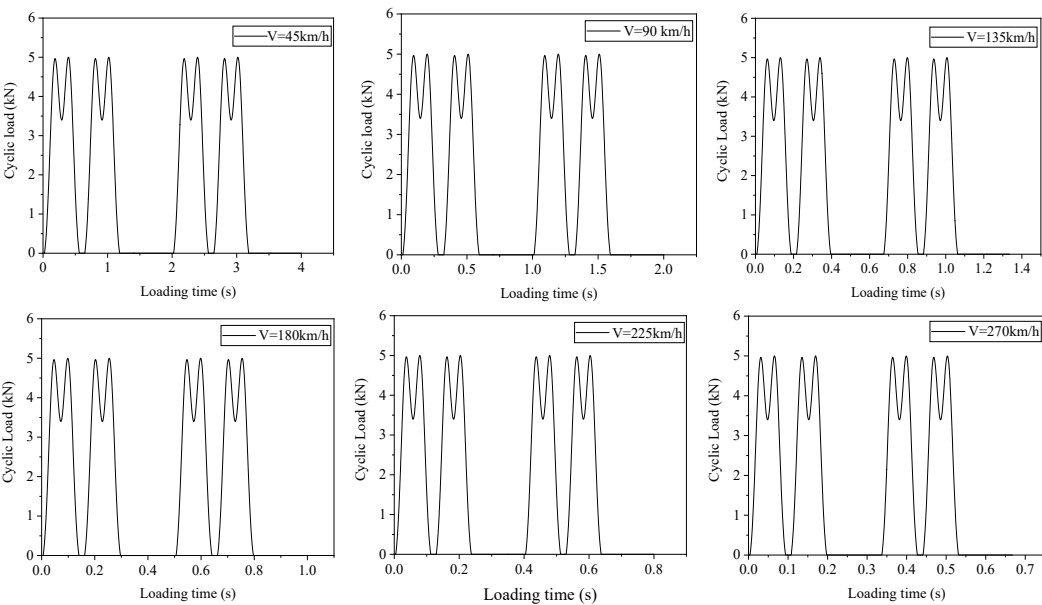

**Figure 5.** Actuator loading time–history curves at a reduced scale for different train speeds.

A dynamic loading hydraulic servo control and digital data acquisition system (composed of a real-time data display system, servo loading control system, Moog-controller, data acquisition system) were established to realize the dynamic loading test (see Figure 6). The computer program used in the servo loading control system is capable of coordinating the actuator loading time–history curves. The Moog controller aims to let the actuators work following the given computer program. Meanwhile, the data acquisition system

acquires the related real-time test data accurately. The loading frequency range for this system is 0.1–30 Hz. The highest peak frequency of the M-shaped wave simulated in the model loading test was 30 Hz.

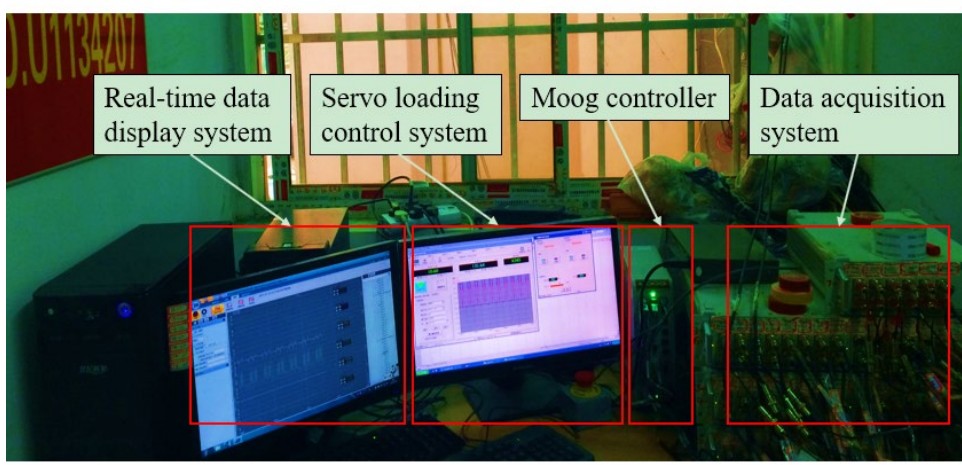

**Figure 6.** Dynamic loading hydraulic servo control and digital data acquisition system.

## 3. Experiment Results and Discussion

In the model tests, train speeds varying from 45 to 270 km/h were simulated through the dynamic loading control system. Various sensors were installed in the physical model of the ballastless railway to monitor the dynamic response of the track structure and piled raft foundation. Vibration sensors were placed on the surface of the track structure, roadbed, embankment, and subsoil in the transversal direction from the track center (see Figure 2a). Velocity sensors V1, V2, V3, V4, and V12 were fixed on the surface of the track slab, roadbed, road shoulder, slop toe, and ground surface. Velocity sensors V6, V7, V8, V9, V10, and V11 were embedded in the top, middle side, and bottom position of the soft soil to record the dynamic response of the subsoil.

### 3.1. Velocity Response of the Track and Pied Raft Foundation

Figures 7 and 8 refer to the time history and frequency contents of the vibration velocities at the track slab V1, roadbed V2, road shoulder V3, and slop toe V4, for the low speed (e.g., 90 km/h) and high speed (e.g., 270 km/h), respectively. Figure 7 (v = 90 km/h) shows the cycle time $T_3 = 1$ s, representing the time during which one carriage passes the monitoring point. The peak value of the vibration velocity varies with the moving train's axle loads. The maximum values of the track slab, roadbed, road shoulder, and slop toe were measured as 12.5, 1.58, 1.25, 0.84 mm/s, respectively. It is obvious that the peak value decreases quickly from the track slab to the roadbed with an amplitude of 87.4%, and then slows down gradually along the roadbed to the slope toe. For the case of v = 270 km/h (Figure 7b,d,f,h), the cycle time is $T_3 = 0.333$ s, representing one carriage passing time. At the same locations, V1, V2, V3, and V4, maximum vibration velocities were obtained as 37.5, 4.22, 3.12, 3.45 mm/s, respectively. The peak vibration velocity decreases quickly by 88.8% from the track slab to the roadbed, and then slows down gradually along the roadbed to the slope toe. There is a certain fluctuation of the velocity at the structural break of the raft and embankment. The structural break has an influence on the vibration response of the track, embankment, and pile–raft foundation system. The variation pattern of the vibration velocity corresponds to loading from the bogies. The vibration velocity of the roadbed, shoulder, and slope toe follows a pattern similar to that of the track slab. The difference is that the vibration velocity at a train speed of 270 km/h is higher than that at a train speed of 90 km/h. The vibration velocity at each location can reflect the applied M-wave load and the vibration state of the pile–raft foundation well. The intensity of the vibration velocity curves at each location under the high-speed load is clearly higher

than that under the low-speed load, including the amplitude, similarity, and fluctuations of the intensity, which are associated with the increase of the train speed.

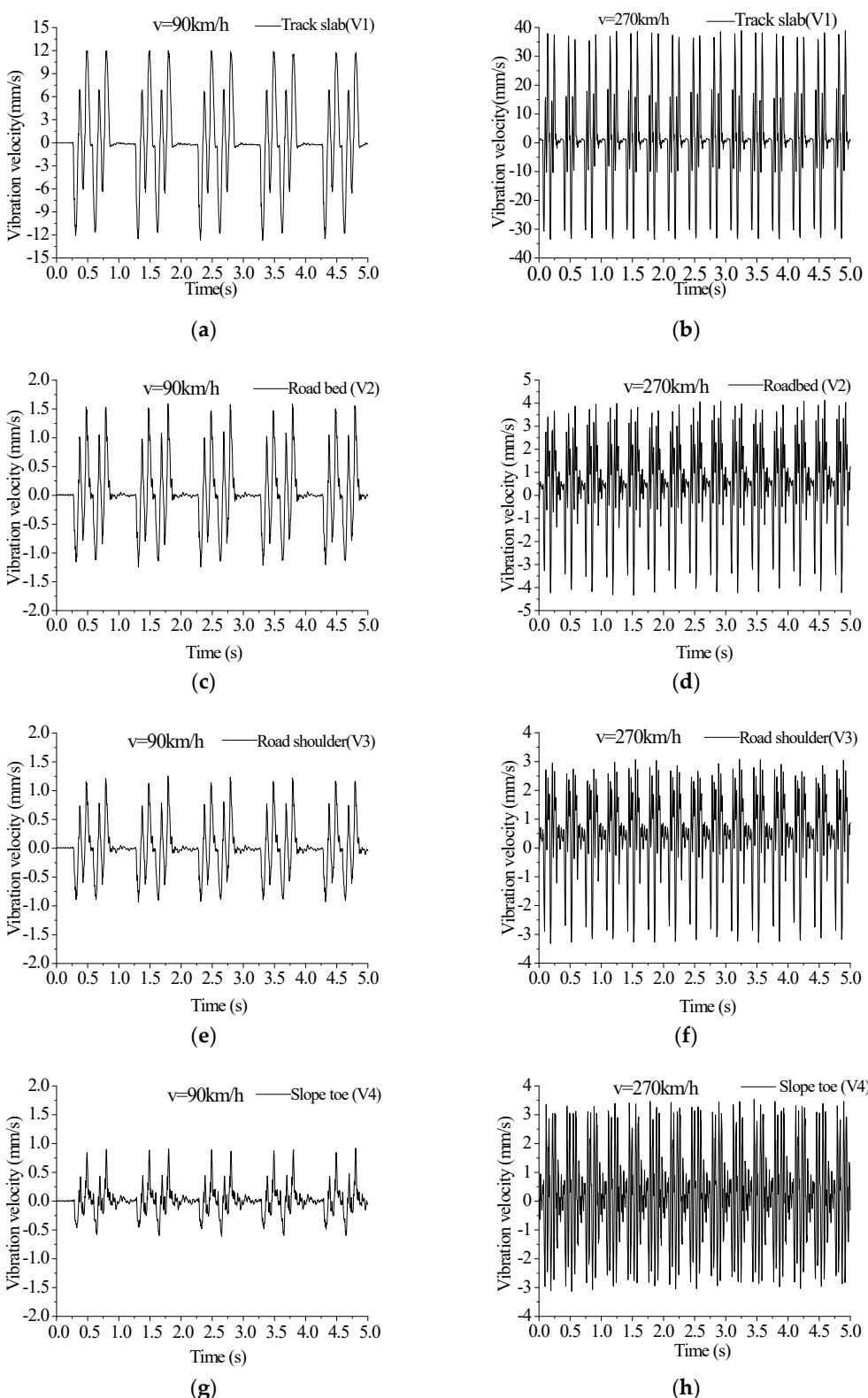

**Figure 7.** Time history curves of the vibration velocities at the track slab, roadbed, road shoulder, and slope toe for different train speeds. (**a**) Track slab (V1, v = 90 km/h), (**b**) Track slab (V1, v = 270 km/h), (**c**) Roadbed (V2, v = 90 km/h), (**d**) Roadbed (V2, v = 270 km/h), (**e**) Road shoulder (V3, v = 90 km/h), (**f**) Road shoulder (V3, v = 270 km/h), (**g**) Slop toe (V4, v = 90 km/h), (**h**) Slop toe (V4, v = 270 km/h).

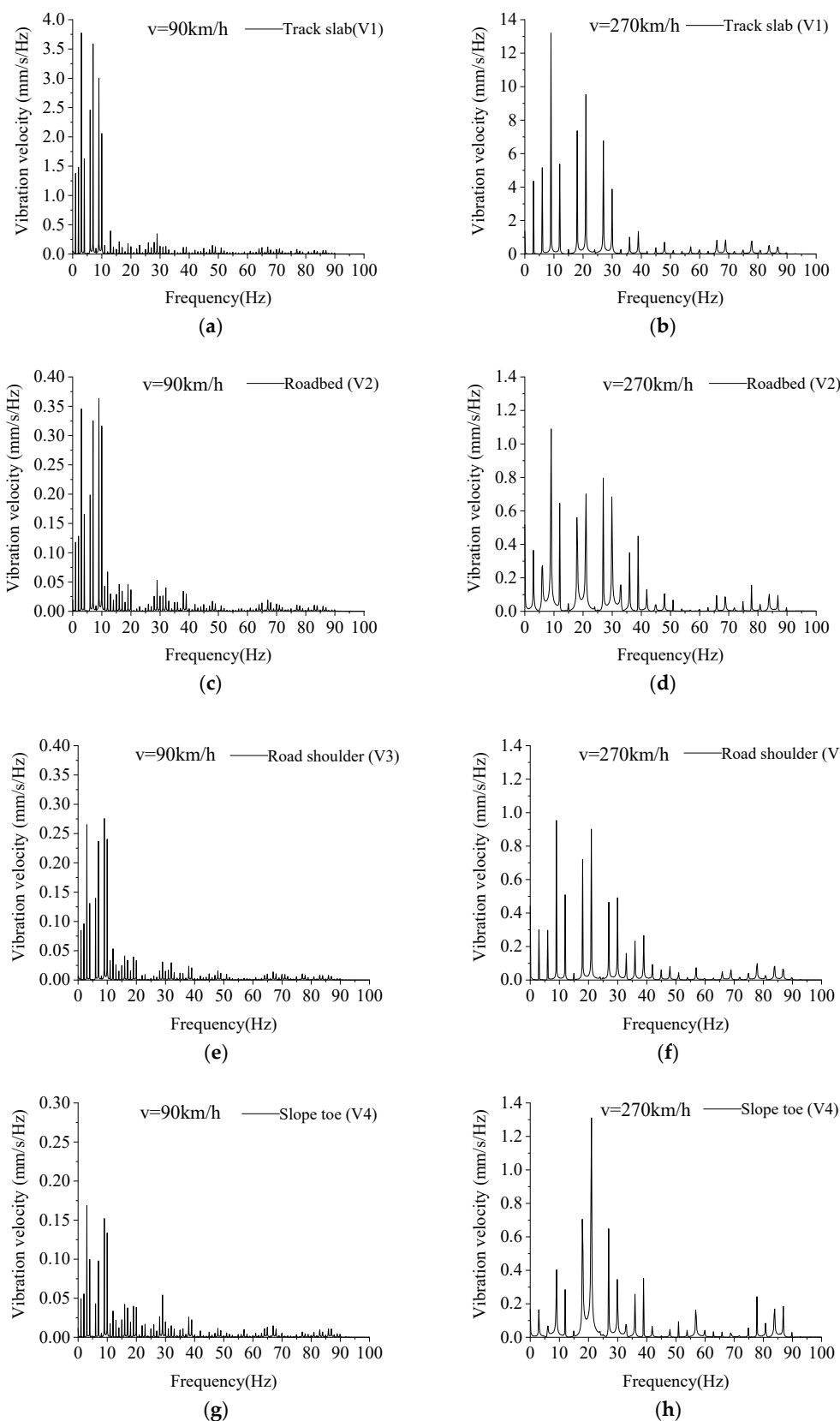

**Figure 8.** Frequency spectrum analysis of vibration velocities at the track slab, roadbed, road shoulder, and slope toe for different train speeds. (**a**) Track slab (V1, v = 90 km/h), (**b**) Track slab (V1, v = 270 km/h), (**c**) Roadbed (V2, v = 90 km/h), (**d**) Roadbed (V2, v = 270 km/h), (**e**) Road shoulder (V2, v = 90 km/h), (**f**) Road shoulder (V3, v = 270 km/h), (**g**) Slop toe (V4, v = 90 km/h), (**h**) Slop toe (V4, v = 270 km/h).

In the case of low speed (90 km/h) and high speed (270 km/h), the peak value of the track slab increases three-fold. With the vibration velocity moving to the roadbed and the shoulder in turn, the proportion value of the peak velocity at high speed and low speed decreases gradually. The velocity attenuation is relatively large from the track slab to the roadbed but is relatively small from the roadbed to the slope toe.

The frequency spectrum curves of the vibration velocity for the low and high train speeds are compared in Figure 8. For the case of $v = 90$ km/h, typical characteristic frequencies of the vibration velocity are 1, 3, and 9 Hz, corresponding to the passing frequency of the train carriage geometry ($l_c = 25$ m), bogie ($l_{ab} = 7.5$ m), and axle distance ($l_{wb} = 2.5$ m), respectively. The frequency spectral pattern also satisfies the case of $v = 270$ km/h in Figure 8b,d,f,h, with typical characteristic frequencies of 3, 9, and 27 Hz. These frequencies have a harmonic multiple relationship with each other. Notice that the typical peaks of the frequency amplitude are also related to the geometric parameters of the trains (bogie, carriage, axle), and the vibration speed spectrum at high speed is about 3 times that at low speed, which is consistent with the above analysis of the relationship between the vibration velocity and train speed. At the same time, the frequency corresponding to the peak value of the vibration velocity spectrum also gradually moves towards the medium frequency with the increase of the train speed. Besides, the corresponding peak value and frequency of the velocity spectrum at each position are also different along the subgrade, which can be attributed to the propagation attenuation process of the vibration velocity.

### 3.2. Internal Vibration Velocity Response of the Track and Pile–Raft Foundation

Figure 9 illustrates the variation of the vibration velocities at different locations (V5, V6, V7, V11) with the time history. It can be found in Figures 7 and 8 that the velocity responses are consistent with the characteristics of the dynamic loading curve for both low and high train speeds, and similar results can also be found in Figure 9. For the case of $v = 90$ km/h, the maximum vibration velocity at location V5 (0.15 m under the roadbed surface) is 0.6 mm/s, which decreases by 50% compared with that at V2 (on the roadbed surface). The maximum velocity values at the raft (V6), cushion (V7), and at the bottom of the subsoil (V11) are 0.58, 0.48, 0.08 mm/s, respectively. The attenuation ranges of the velocity from the roadbed bottom to the subsoil bottom (i.e., V6 to V11) reach 86.7%. In contrast, the amplitude and intensity of the velocity are increased for the case of $v = 270$ km/h. The vibration velocity amplitudes at V5, V6, V7, and V11 are 4.2, 2.55, 1.58, and 0.45 mm/s, which are 7, 4.4, 3.3, and 5.6 times more than those at the corresponding position of $v = 90$ km/h. The attenuation ratio of the velocity from the roadbed bottom V5 to the subsoil V11 is 82.4%. The vibration velocity propagates down along the embankment and attenuates gradually. Vibration isolation and attenuation phenomena occur in the embankment and pile–raft foundation. The vibration waves move along the subsoil, and decrease to a very low level at the subsoil, which is much lower than that at the roadbed surface.

The frequency contents of the vibration velocities at the road, raft, and subsoil are shown in Figure 10. For $v = 90$ km/h, the peak values appear at a frequency point below 10 Hz, including 1, 3, and 9 Hz, reflecting the characteristic frequencies of the geometry of the train compartment. The peak values of the vibration velocity frequency spectrum are distributed in low-frequency and medium-frequency regions, respectively. In Figure 10a, there is a small amplitude of vibration energy in the 27~40 Hz region of the roadbed, and the vibration wave is enhanced in this frequency band, indicating that the characteristic frequency of the vibration of the roadbed packing is distributed in this region. In Figure 10b, for $v = 270$ km/h, information of the frequency contents shows peaks at about 3, 9, and 27 Hz. Compared with the low speed, the peak points of the vibration speed spectrum gradually move to the medium-frequency area, and increase with the increase of the train speed. Larger velocity spectrum peaks occur in the range of 27~40 Hz, which is 8 times more than that at low speed, and the ratio of vibration energy is higher than that at low speed.

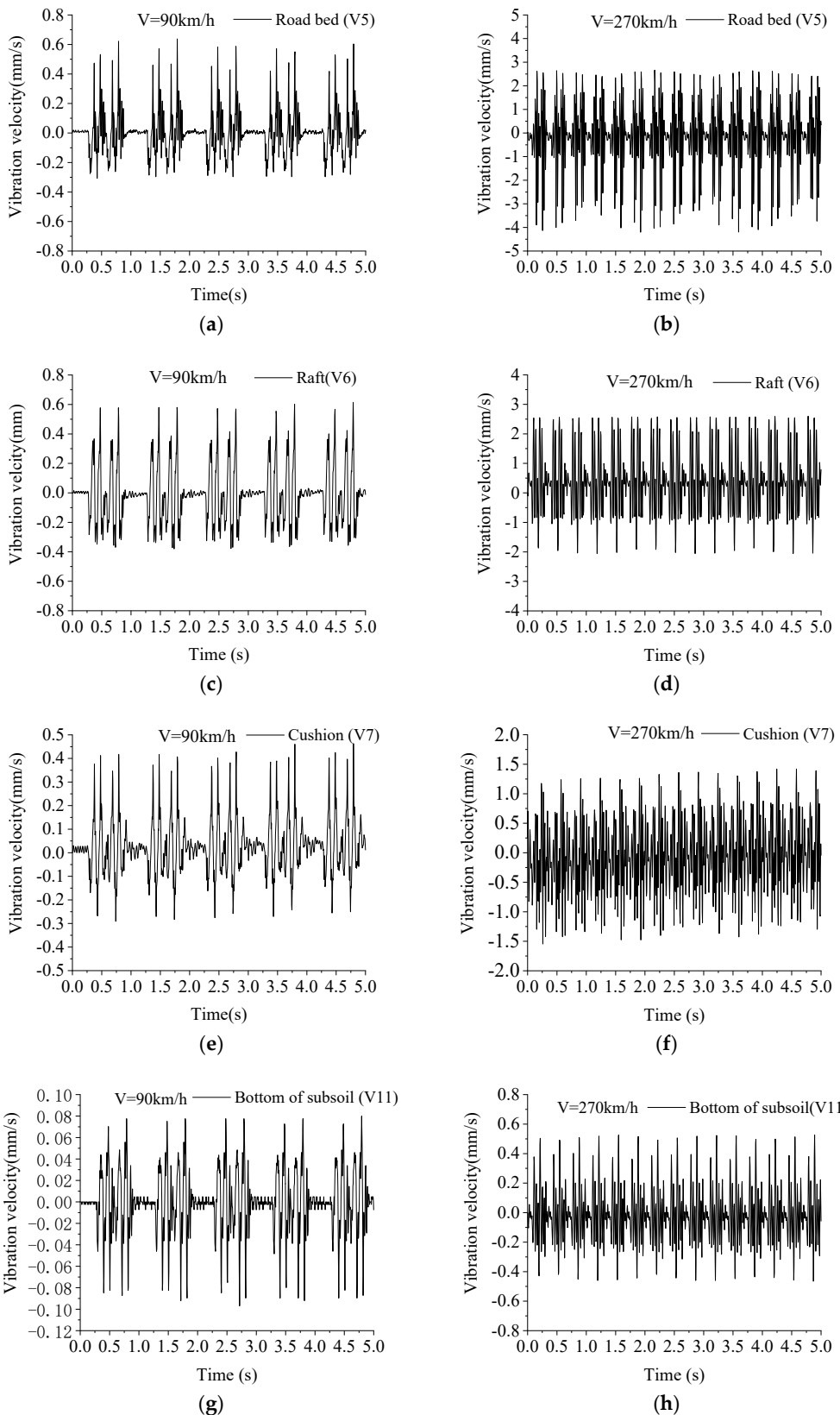

**Figure 9.** Time history curves of vibration velocities at different locations of the pile–raft foundation. (**a**) Roadbed (V5, v = 90 km/h), (**b**) Roadbed (V5, v = 270 km/h), (**c**) Raft (V6, v = 90 km/h), (**d**) Raft (V6, v = 270 km/h), (**e**) Cushion (V7, v = 90 km/h), (**f**) Cushion (V7, v = 270km/h), (**g**) Bottom of subsoil (V11, v = 90 km/h), (**h**) Bottom of subsoil (V11, v = 270 km/h).

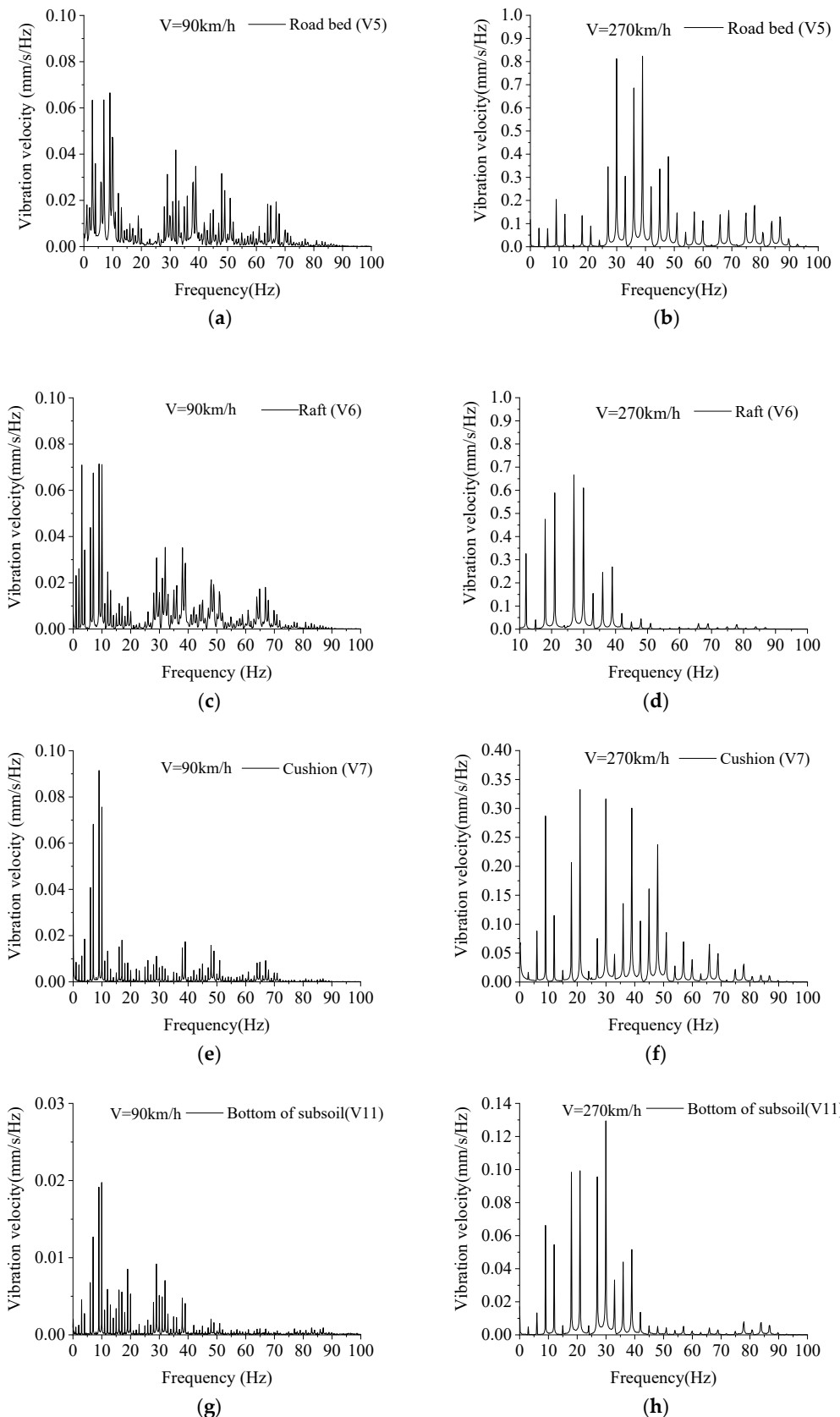

**Figure 10.** Frequency spectrum analysis of vibration velocities at different locations of a track embankment pile–raft foundation. (**a**) Roadbed (V5, v = 90 km/h), (**b**) Roadbed (V5, v = 270 km/h), (**c**) Raft (V6, v = 90 km/h), (**d**) Raft (V6, v = 270 km/h), (**e**) Cushion (V7, v = 90 km/h), (**f**) Cushion (V7, v = 270 km/h), (**g**) Bottom of subsoil (V11, v = 90 km/h), (**h**) Bottom of subsoil (V11, v = 270 km/h).

For low speed and high speed, the frequency contents of the vibration velocity are mainly distributed in the low and medium region, the peak values of which respectively correspond to the passing frequency of the train carriage geometry ($l_c$ = 25 m), bogie ($l_{ab}$ = 7.5 m), and axle distance ($l_{wb}$ = 2.5 m), reflecting the characteristic frequencies of the train compartment, adjacent bogie, and wheel load passing through. The peak spectrum at high speed is significantly higher than that at low speed.

A similar pattern can be found in Figure 10c–h, where the difference is the attenuation amplitude of the velocity spectrum at different locations. From the roadbed and the raft to the bottom of the subsoil, the distribution area and peak frequencies of the amplitude spectrum of the vibration velocity changes with the depth. The peak values of the frequency spectrum gradually decrease and move from the medium- to low-frequency energy distribution area along the vertical depth, and the medium-frequency parts of the peak values and spectrum energy decay gradually. This phenomenon is more obvious when the train is moving at high speed. The energy distribution of the vibration velocity spectrum is gradually closer to the medium and low frequency, which is also greatly related to the damping attenuation of the embankment and soil foundation. The high-frequency part is filtered out, leaving the medium- and low-frequency part with a lower amplitude, thus reducing the vibration response of the foundation soil.

### 3.3. Influence of Train Speed on Vibration Velocity

Figure 11 refers to variations of the pile–raft foundation's velocity with the train speed. It can be seen that the present vibration velocity tends to increase with the increasing train speed. The increasing rate at the track slab, roadbed, road shoulder, slope toe, and the ground soil is about 1.38, 0.135, 0.115, 0.133, and 0.104 mm/s per 10 km/h. Note in Figure 11a that the vibration velocity at the track slab (i.e., V1) is sensitive to the train speed compared to that at the road bed, road shoulder, slope toe, and ground surface (i.e., V2, V3, V4, and V12), respectively, due to the higher stiffness of the structure. Similar variations can be found in Bian's study [26]. As the highest peak frequency of the M-shaped wave simulated in the model loading test was 30 Hz, the maximum train speed in the model test was 270 km/h, and the dynamic response analysis of the track embankment and pile–raft foundation at higher speeds can be achieved by numerical simulation.

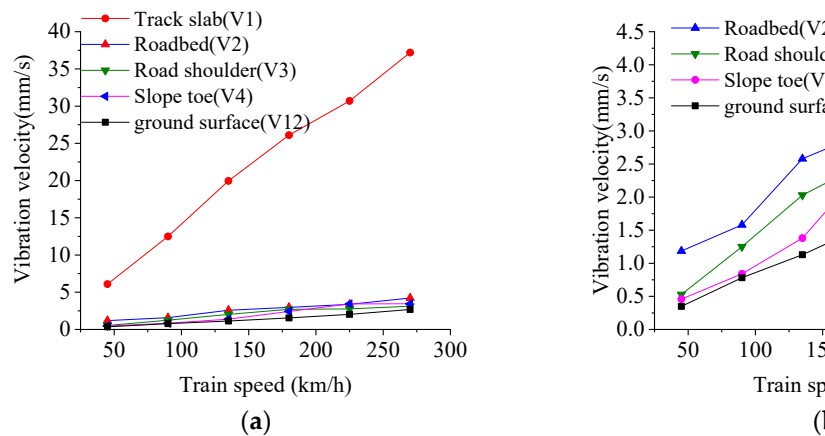

**Figure 11.** Relationship between the vibration velocity and train speeds in the model test. (**a**) V1, V2, V3, V4, V12, (**b**) V2, V3, V4, V12.

The relationship between the vibration velocity and train speed at the locations V5, V6, V7, and V11 is shown in Figure 12. It can be seen that that the vibration velocity response of the roadbed, raft, cushion, and bottom of the subsoil increases with the increase of the train speed. The growth rate is 0.178, 0.098, 0.061, and 0.0174 mm/s per 10 km/h, respectively. Additionally, from Figure 12, it can be seen that the vibration velocity's growth rate of

the roadbed (V5) is 0.44 mm/s per 10 km/h when the train speed is increased from 225 to 270 km/h, which is higher than the others, implying that the vibration response at the track and roadbed is more sensitive to the high train speed. As expected, the velocity's growth rate of internal vibration is comparably lower than that of the surface. The upper vibration waves are reflected and attenuated by the isolation of the raft structure, resulting in a lesser vibration response of the subsoil under the raft. The sensitivity of the vibration velocity response of the foundation soil to the train speed is much lower than that of the upper structures, such as the track slab, roadbed, and embankment.

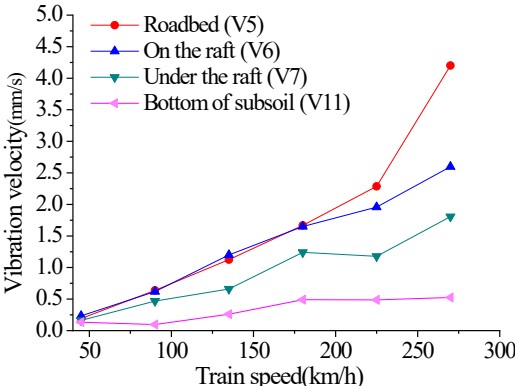

**Figure 12.** Relationship between the vibration velocity and train speed at the locations V5, V6, V7, and V11.

## 4. Vibration Attenuation Pattern of the Track Embankment and Pile–Raft Foundation

Figure 13 shows values of the peak velocity for different locations (V1, V2, V3, V4) under different train speeds. The vibration response level and attenuation are different for different train speeds. The data fitting results are shown in Table 6 and Figure 13. The data curves of the vibration velocity peaks under the different train speeds were fitted using the following equations:

$$V_{90km/h} = 522e^{\frac{-x}{0.065}} + 1.04 \tag{4}$$

$$V_{180km/h} = 3890e^{\frac{-x}{0.048}} + 2.56 \tag{5}$$

$$V_{225m/h} = 8781e^{\frac{-x}{0.043}} + 3.09 \tag{6}$$

$$V_{270km/h} = 3080e^{\frac{-x}{0.055}} + 3.28 \tag{7}$$

where *V* is the vibration speed (corresponds to y in Table 6, unit, mm) and *x* is the distance up to the track center.

**Table 6.** Data fitting results of peak velocities.

| Model | ExpDec1 | | | |
|---|---|---|---|---|
| Equation | y = A1*exp(−x/t1) + y0 | | | |
| Plot | *v* = 90 km/h | *v* = 180 km/h | *v* = 225 km/h | *v* = 270 km/h |
| y0 | 1.04 | 2.56 | 3.09 | 3.28 |
| A1 | 522.91 | 3890.77 | 8781.30 | 3080.76 |
| t1 | 0.065 | 0.048 | 0.0433 | 0.0555 |
| Reduced Chi-Sqr | 0.083 | 0.041 | 0.224 | 0.0548 |
| RSquare (COD) | 0.9991 | 0.9999 | 0.9996 | 0.9999 |
| Adjusted RSquare | 0.9973 | 0.9996 | 0.9988 | 0.9998 |

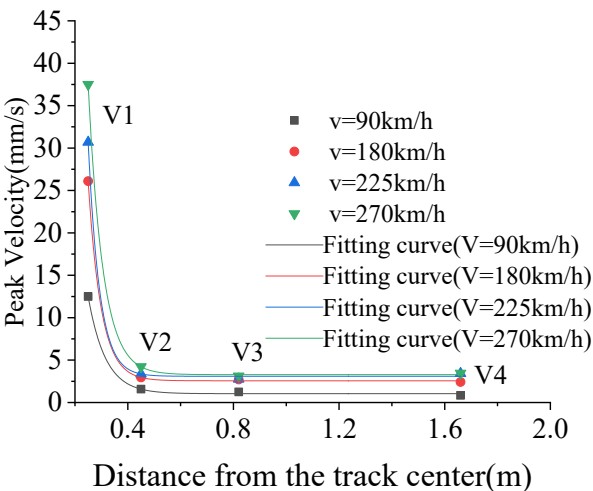

**Figure 13.** Distribution of the vibration velocity in the horizontal direction from the track center.

The fitting curves shows that the vibration velocity attenuation follows the exponential curve distribution from the low to high train speed. The peak velocity depends significantly on the distance, decreasing quickly to a distance of up to 0.45 m and slowing down from 0.45 to 1.66 m. The peak velocity increases with the train speed. Further prediction and analysis of the vibration velocity response of the pile–raft composite foundation can be performed on high-speed train loads in soft soil areas. The vibration velocity attenuation rates for the track embankment and pile–raft foundation in the horizontal direction are shown in Table 7. For $v = 90$ km/h, $v = 180$ km/h, $v = 225$ km/h, and $v = 270$ km/h, the amplitude of the vibration velocity from the track slab (V1) to the roadbed (V2) are attenuated by 87.36%, 88.66%, 89.02%, and 88.75%, respectively. In turn, the attenuation rates from the roadbed to the shoulder are 5.92%, 2.07%, −0.20%, and 2.05%, respectively. This indicates that the attenuation rates for different train speeds are relatively close, and vibration velocity attenuation mainly occurs from the track slab to the roadbed. Most vibration attenuation is completed in the process of vibration propagation from the track slab to the road bed. The velocity attenuation tendencies in the ballast, embankment, and ground are different due to their dynamic characteristics.

**Table 7.** Vibration velocity attenuation rate in the horizontal direction.

| Train Speed | $v = 90$ km/h | $v = 180$ km/h | $v = 225$ km/h | $v = 270$ km/h |
|:---:|:---:|:---:|:---:|:---:|
| **Location** | **Rate** | **Rate** | **Rate** | **Rate** |
| V1 | 0 | 0 | 0 | 0 |
| V2 | 87.36% | 88.66% | 89.02% | 88.75% |
| V3 | 90.00% | 89.62% | 91.01% | 91.68% |
| V4 | 93.28% | 90.73% | 88.83% | 90.80% |

Figure 14 shows the distribution of the vibration velocity with the depth from the roadbed surface in the middle cross-section of the track embankment and subsoil foundation. The amplitudes of the vibration velocity at the roadbed are 1.58 and 4.22 mm/s at train speeds of 90 and 270 km/h, respectively, which is reduced by 95% and 93.2% from the track slab V1. The existence of stiffness of the raft has a certain effect on the transmission, reflection, and superposition of the vibration wave inside the embankment, and results in slower attenuation of the vibration velocities. The amplitudes of the vibration velocity at the bottom of the subsoil are 0.08 and 0.526 mm/s, which are much smaller than those at the roadbed V1. This is because the pile–raft structure supports most of the vibration loads, and blocks the transmission of the vibration vibrating waves, resulting in comparably lower vibration vibrating velocities. The effect of the moving train load on the vibration velocities

of the soft soil foundation is very small. The pile–raft structure composite foundation over soft soil can effectively reduce the level of the vibration response induced by high train loads.

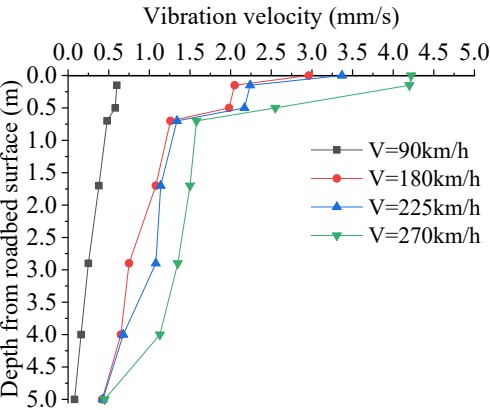

**Figure 14.** Distribution of the vibration velocity amplitude along the vertical direction from the roadbed.

Notice from Figure 14 that the velocities decrease along the depth from the roadbed to raft, and have a vibration enhancement region (from the raft to a 1 m depth of the foundation soil). It shows that the vibration wave forms a small strengthened area at 1 m of the foundation soil, and then decreases gradually along the foundation. For the case of $v = 270$ km/h, the maximum velocity at the bottom of the subsoil is 0.45 mm/s, which is much lower than that at the embankment.

The vibration attenuation rates of the velocity along the depth from the roadbed are shown in Table 8. The vibration attenuation rates of the velocity from the roadbed to the bottom of the subsoil are 94.94%, 85.84%, 87.33%, and 89.34% at train speeds of 90, 180, 225, and 270 km/h, respectively. The vibration velocity decreases at much faster rates within the first 0.7 m of depth from the roadbed surface, and increases within 1 m of depth, and then the rates slow down. With the increase of the depth, the vibration velocities decrease and increase along the embankment–pile–raft foundation. Besides, the increase of the train speeds causes a certain increase of the velocity at different locations of the embankment–pile–raft foundation in soft soil.

**Table 8.** Vibration attenuation rates of velocity along depth.

| Train Speed | $v$ = 90 km/h | $v$ = 180 km/h | $v$ = 225 km/h | $v$ = 270 km/h |
|:---:|:---:|:---:|:---:|:---:|
| **Depth/m** | **Rate** | **Rate** | **Rate** | **Rate** |
| 0.15 (V5) | 62.03% | 30.74% | 33.53% | 0.47% |
| 0.5 (V6) | 63.29% | 33.11% | 35.61% | 39.57% |
| 0.7 (V7) | 69.62% | 57.43% | 60.24% | 62.56% |
| 1.7 (V8) | 75.95% | 63.51% | 66.17% | 64.45% |
| 2.9 (V9) | 84.18% | 74.66% | 67.95% | 68.01% |
| 4 (V10) | 89.87% | 78.04% | 79.82% | 73.22% |
| 5 (V11) | 94.94% | 85.84% | 87.33% | 89.34% |

## 5. Three-Dimensional Dynamic Finite Element Analyses
*Model Development*

The three-dimensional elastic constitutive finite element model illustrated in Figure 15 was used to calculate the vertical vibration velocities of the track–embankment–pile raft foundation. The elastic dynamic responses for different structure layers were calculated by using dynamic simulations at a reduced scale under train loads. The total length, width, and height of the model was 5, 4, and 5.3 m, respectively. The translation in the z-direction

was restrained on the symmetry boundary. A fixed boundary was applied to the bottom of the model. Infinite elements based on the previous work by Lysmer and Kuhlemeyer [34] and Kouroussis et al. [35] were used on the $X$ and $Z$ direction boundaries to represent the infinite boundary condition to absorb $S$ and $P$ waves. As for the other four boundaries, infinite elements were applied to represent the infinite boundary condition. The material parameters applied to the infinite elements were the same as their adjacent solid elements. The material parameters for the different structures layers used in the finite element model are listed in Table 9. The railway track, track slab, concrete base, roadbed, concrete raft, supporting layer, and pile were assumed to be liner elastic based on their high stiffness. The embankment, gravel, and silty soil were assumed to be viscoelastic. Rayleigh damping was used to represent the energy-dissipating mechanisms for the viscoelastic material in the simulation model, in which the mass and stiffness proportional damping coefficients were selected to provide slight damping ratios of 2–4% in the frequency range of 3–50 Hz [36–38]. The mass and stiffness proportional damping coefficients of embankment, gravel, and subsoil were selected and are shown in Table 9.

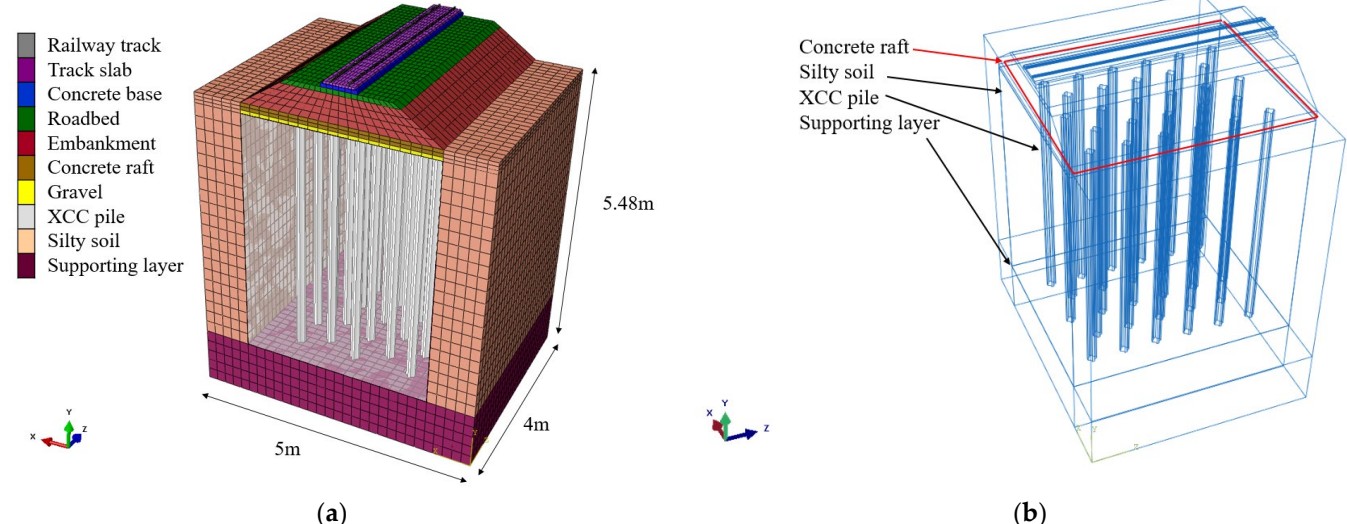

**Figure 15.** Finite element model of the pile–raft composite foundation. (**a**) finite element mesh, (**b**) wireframe of the model.

**Table 9.** Material parameters of the dynamic loading test.

| Structure Layer | Thickness (m) | Elastic Modulus $E$ (MPa) | Mass Density (kg/m³) | Poisson's Ratio | Damping Coefficient ($\alpha$, $\beta$) |
|---|---|---|---|---|---|
| Railway track | 0.03 | 210,000 | 78 | 0.2 | - |
| Track slab | 0.04 | 30,000 | 25 | 0.2 | - |
| Concrete base | 0.06 | 30,000 | 25 | 0.2 | - |
| Roadbed | 0.08 | 200 | 22 | 0.3 | - |
| Embankment | 0.46 | 120 | 20 | 0.3 | 1.2, 0.0004 |
| Concrete raft | 0.12 | 30,000 | 25 | 0.2 | - |
| Gravel | 0.06 | 200 | 20 | 0.3 | 1.2, 0.0004 |
| Silty soil | 4.3 | 20 | 15.6 | 0.35 | 1.66, 0.0033 |
| Supporting layer | 1.0 | 50 | 20 | 0.3 | - |
| Pile | 4.3 | 30,000 | 25 | 0.2 | - |

In Figure 16, the equivalent moving M-shaped loading wave was used to simulate the train moving load by applying the load on the unit point in the numerical simulation, of which the time and frequency domain features were consistent with the loading waves in the model test. The loading wave includes the passing frequency of the train carriage geometry ($l_c$ = 25 m), bogie ($l_{ab}$ = 7.5 m), and axle distance ($l_{wb}$ = 2.5 m). The numerical

simulation results were calculated and compared with the model test results to verify the reliability and practicability of the numerical simulation method.

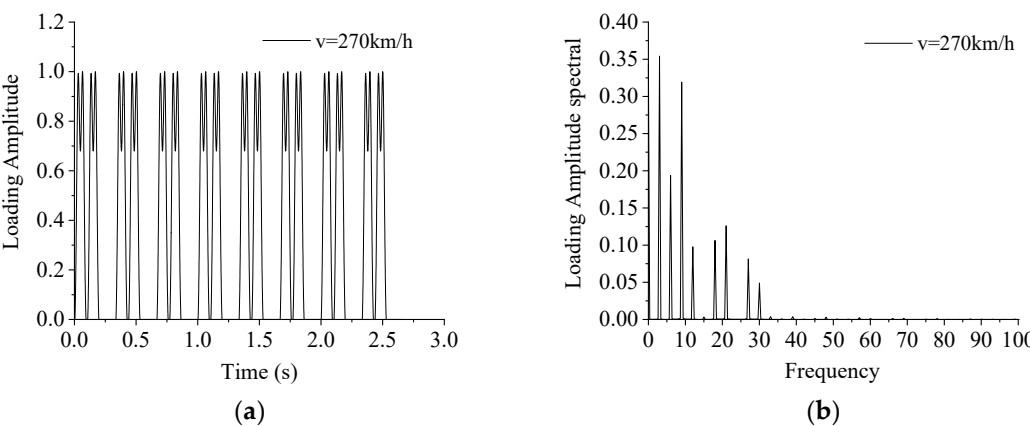

**Figure 16.** Dynamic loading wave (v = 270 km/). (**a**) waveform of dynamic loading wave, (**b**) frequency distribution of dynamic loading wave.

## 6. FE Results and Analysis

### 6.1. Comparison of the FE with the Test Results in the Time and Frequency Domain at Different Locations

In this section, the FE results are compared with the test results. For train speed $v$ = 270 km/h, the vibration velocities at the track slab, roadbed, embankment, and pile–raft foundation were calculated in the numerical model.

Figure 17b,d,f,h,j show the frequency domain analysis results corresponding to the vibration velocity time histories in Figure 16a,c,e,g,i, respectively. Figure 17a shows the time history of the vibration velocity at the track slab induced by the train moving at a speed of 270 km/h. The amplitudes of the vibration velocities are 41.48 mm/s in the FE simulation data and 32 mm/s in the model test data. In Figure 17b, the frequency contents of the vibration velocity for the model test and FE simulation present the same dominant frequencies within certain frequency ranges. Most of the spectral energy of the vibration velocity is concentrated at the frequency of about 3, 9, and 27 Hz. These three dominant frequencies correspond to the passing frequency of the train carriage geometry ($l_c$ = 25 m), bogie ($l_{ab}$ = 7.5 m), and axle distance ($l_{wb}$ = 2.5 m), reflecting the characteristic frequencies of the train compartment.

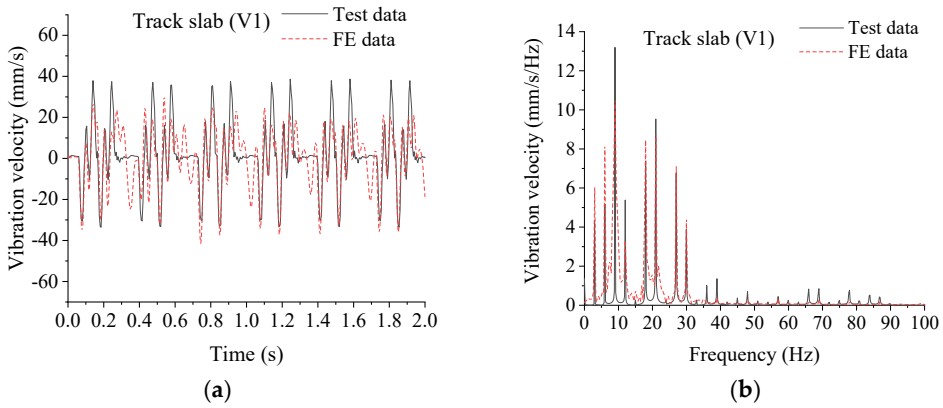

**Figure 17.** *Cont.*

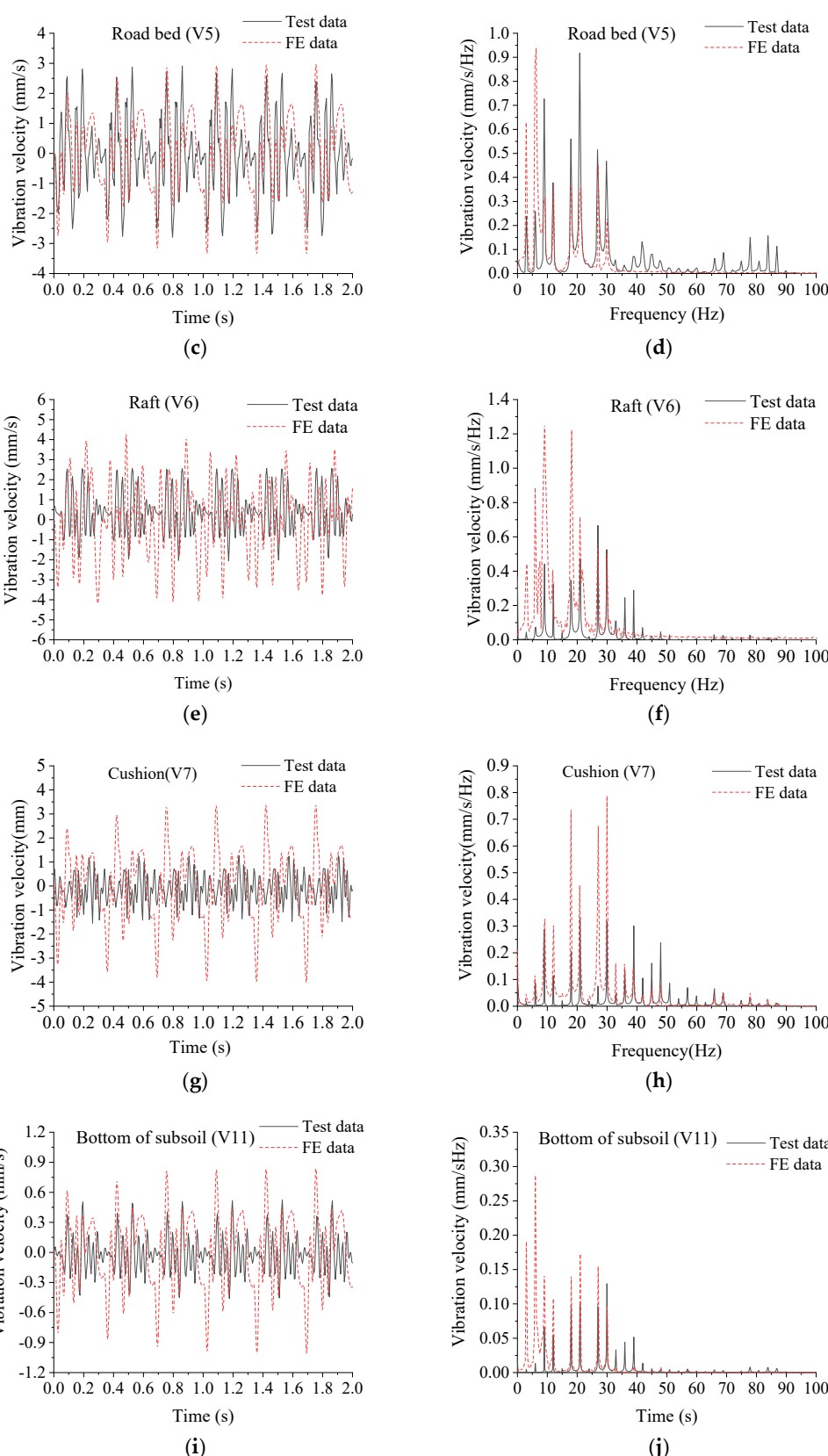

**Figure 17.** Comparisons of the vibration velocity between the FE and test results for train speed v = 270 km/h. (**a**) Track slab (V1, time domain), (**b**) Track slab (V1, frequency domain), (**c**) Roadbed (V5, time domain), (**d**) Roadbed (V5, frequency domain), (**e**) Raft (time domain), (**f**) Raft (frequency domain), (**g**) Cushion (time domain), (**h**) Cushion (frequency domain), (**i**) Bottom of subsoil (time domain), (**j**) Bottom of subsoil (frequency domain).

In Figure 17c,d, the FE results and model test results still agree with each other in the time and frequency domain. The amplitudes of the vibration velocities are 3.35 mm/s in the FE simulation data and 2.72 mm/s in the model test data. The spectral energy of the vibration velocity in the model test is lower than that in the FE simulation at 3 and 6 Hz. It shows that the vibration attenuation of the model test is higher than that of the numerical simulation in some frequency ranges.

In Figure 17e,f, the vibration velocities are 4.25 and 2.55 mm/s for the FE data and test data on the raft. Based on the influence of vibration superposition, interference, and attenuation, the vibration velocity energy for the test data displays fast attenuation of the amplitude at low frequency, and is lower than the FE data.

The same law of the vibration response can also be obtained from Figure 17g,h,i,j. The low-frequency part of the frequency contents for the test data attenuates relatively fast. In general, the time history of the vibration velocity obtained from the numerical calculation agrees well with that from the model test regarding the vibration waveform, amplitude, and frequency characteristics. The water content, degree of compaction, and gradation size will have an impact on the vibration response characteristics of the embankment and pile raft foundation.

### 6.2. Vibration Attenuation Pattern of the Pile–Raft Foundation

Figure 18 shows the vibration velocity amplitudes at the track structure and underlying soils in the transversal direction from the track center at three different speeds. In Figure 18a,b, at the top of the track slab, the maximum velocities of the FE simulation at different train speeds are slightly higher than that of the model test. The attenuation law of the vibration velocity from the track center along the embankment is very consistent, showing only a slight difference in the velocity amplitude. The vibration velocity increases with the train speed. As shown in Figure 18b, for the model test, when the train speed is 270 km/h, the increase range of velocity is significantly higher than that at low speeds. The vibration velocity attenuation rate can reach 90 % from the track slab to the top of the roadbed. It is shown that the vibration intensity of the upper structure of the track slab is large, and decreases quickly to lower than 5 mm/s in the roadbed while the vibration velocity of the lower foundation bed and the embankment structure decreases rapidly to less than 5 mm/s. The influence of the vibration damping attenuation of the roadbed and embankment under different train speeds is consistent, which further verifies the accuracy of the numerical simulation method. The FE numerical simulation results are close to the model test results at high speed.

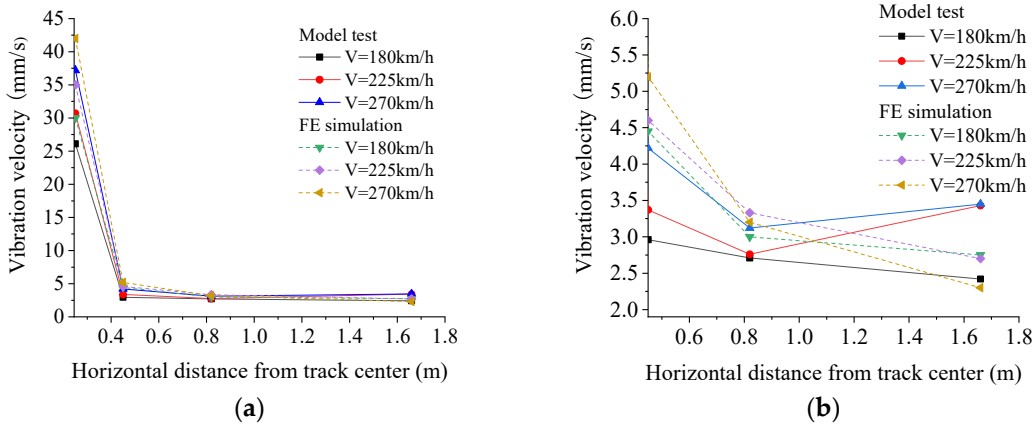

**Figure 18.** Comparison of the vibration velocity attenuation along the horizontal direction. (**a**) V1–V4, (**b**) V2–V4.

Figure 19 shows the distribution of the vibration velocity along the depth from the roadbed in the middle cross-section, where velocity sensors were located under the track center. The peak value obtained from the FE simulation is slightly larger than that from the

model test, and the velocity distribution and attenuation law obtained by the two methods are basically consistent. This is related to the consistency between the parameters of the model test material and the FE simulation. As mentioned earlier, the water content, degree of compaction, and gradation size impact on the vibration characteristic of the embankment and pile–raft foundation, thus resulting in a certain increase of the damping coefficient, and leading to vibration attenuation. For the model test and FE simulation, the velocities decrease along the depth from the roadbed to the raft, and have a vibration enhancement region from the raft to the 1 m depth of the foundation soil. It shows that the vibration wave forms a small area with a vibration strengthening effect at 1 m of the foundation soil, and then decreases gradually along the foundation. For $v$ = 270 km/h, the maximum velocity at the bottom of the subsoil is 0.45 and 1.57 mm/s, which is much lower than that at the embankment.

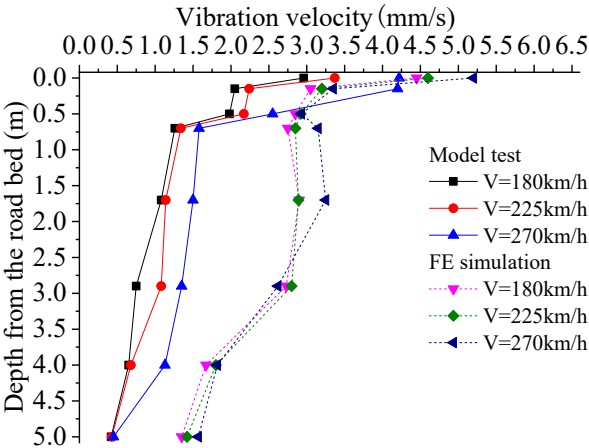

**Figure 19.** Distribution of the vibration velocity along the depth from the roadbed.

The vibration attenuation rates of the velocity along the depth from the roadbed for the model test and FE simulation at different train speeds were calculated and are shown in Table 10. The vibration attenuation rates of the velocity from the roadbed to the bottom of the subsoil are 85.84% and 69.66% at a train speed of 180 km/h, 87.33% and 69.13% at a train speed of 225 km/h, and 89.34% and 69.81% at a train speed of 270 km/h, respectively. The vibration velocities decrease at much faster rates within the first 0.7 m of depth from the roadbed surface, and increase within the 1 m depth, and then the rates slow down. With the increase of the depth, the vibration velocities decrease and increase along the embankment–pile–raft foundation. Besides, the increase of the train speeds causes a certain increase of the velocity at different locations of the embankment and XCC pile–raft foundation.

**Table 10.** Vibration attenuation ratio of the velocity along depth.

| Train Speed | $v$ = 180 km/h | | $v$ = 225 km/h | | $v$ = 270 km/h | |
|---|---|---|---|---|---|---|
| Depth/m | Model Test | FE Data | Model Test | FE Data | Model Test | FE Data |
| 0.15 | 30.74% | 31.46% | 33.53% | 30.43% | 0.47% | 35.58% |
| 0.5 | 33.11% | 35.96% | 35.61% | 36.52% | 39.57% | 43.65% |
| 0.7 | 57.43% | 38.20% | 60.24% | 38.04% | 62.56% | 39.42% |
| 1.7 | 63.51% | 34.83% | 66.17% | 37.17% | 64.45% | 37.50% |
| 2.9 | 74.66% | 38.65% | 67.95% | 39.13% | 68.01% | 49.62% |
| 4 | 78.04% | 62.47% | 79.82% | 60.87% | 73.22% | 65.00% |
| 5 | 85.84% | 69.66% | 87.33% | 69.13% | 89.34% | 69.81% |

## 7. Conclusions

In this paper, a series of dynamic large-scale model tests and three-dimensional finite element analyses were conducted to investigate the dynamic response of an XCC pile–raft composite foundation for a ballastless high-speed railway under train moving loads. The tests and FE results are presented and were compared regarding the variation of the vibration velocity to analyze the characteristics of the dynamic response, transmission, and attenuation for the XCC pile–raft composite foundation. The main findings can be summarized as follows:

For the model test and FE simulation, the vibration velocity at each location reflects the applied M-wave load and the vibration state of the pile–raft foundation well. The intensity of the vibration velocity curves at each location at high speed is clearly higher than that at low speed, including the amplitude, similarity, and fluctuations of the intensity, which are associated with the increase of the train speed.

The frequency contents of the vibration velocity for the model test and FE simulation present the same dominant frequencies in low and medium regions, the peak value of which respectively corresponds to the passing frequency of the train carriage geometry ($l_c$ = 25 m), bogie ($l_{ab}$ = 7.5 m), and axle distance ($l_{wb}$ = 2.5 m), reflecting the characteristic frequencies of the train compartment, adjacent bogie, and wheel load passing through. The peak spectrum value at high speed is significantly higher than that at low speed.

The peak velocity depends significantly on the distance from the track center in the horizontal direction, mainly decreasing by 89.3% and 87% from the track slab to the roadbed for train speeds of 90 and 270 km/h, respectively. The attenuations of the peak velocity follow the exponential curve distribution from low to high train speed. Most vibration attenuations were completed in the process of vibration propagation from the track slab to the roadbed. The velocity attenuation tendencies in the ballast, embankment, and ground are different due to their dynamic characteristics.

The vibration velocity propagates down along the embankment and attenuates gradually. Vibration isolation and attenuation phenomena occur in the embankment and pile-raft foundation. The velocity distribution and attenuation law obtained from the test and the FE simulation results are basically consistent. The vibration velocities decrease rapidly within the first 0.7 m of depth from the roadbed surface to the raft, and display a vibration enhancement region from the raft to a 1 m depth of the foundation soil. The vibration wave forms a small area with the vibration strengthening effect at 1 m of the foundation soil, and then decreases gradually along the subsoil foundation, to a very low level at the bottom of the subsoil, which is much lower than that at the track slab and roadbed. The pile–raft composite foundation can reduce the vibration level effectively, and improve the safety of trains running in soft soil areas.

**Author Contributions:** Q.F. wrote the manuscript and performed the numerical analyses; J.Y. investigated and reviewed the material parameters required for the numerical and other analyses. All authors have read and agreed to the published version of the manuscript.

**Funding:** This work was funded by the National Natural Science Foundation of China (Grant No. 51908152).

**Institutional Review Board Statement:** Not applicable.

**Informed Consent Statement:** Not applicable.

**Data Availability Statement:** Not applicable.

**Conflicts of Interest:** The authors declare no conflict of interest.

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
