# Peer review of "Experimental and Numerical Study of the Dynamic Response of XCC Pile–Raft Foundation under High-Speed Train Loads"

_applsci, doi:10.3390/app11199260_

Round 1
Reviewer 1 Report
The manuscript is overall interesting. However, some issues are found throughout the paper. Therefore, according to this Reviewer, a moderate revision would be necessary before the paper can be further considered for possible publication in Applied Sciences. All details are summed up in the following.
Required changes:
- The title seems to be too long. A shorter version should be chosen.
- Originality/novelty of the study proposed. This issue is very important and should be better clarified and well highlighted in the text.
- Section 4.1: it is necessary to clearly explain what constitutive models were employed to perform the analysis, giving a proper justification.
- Figure 16: this figure should show the configuration of the piles, which cannot be seen in the current form. Besides, it is necessary to clearly distinguish, using different colors, all the materials indicated in Table 9.
- For the sake of completeness, the following references could be added in the introduction.
SUGGESTED REFERENCES
Achmus, M., and K. Thieken. 2010. “On the behavior of piles in non-cohesive soil under combined horizontal and vertical loading.” Acta Geotech. 5 (3): 199–210. https://doi.org/10.1007/s11440-010-0124-1.
Conte, E.; Pugliese, L.; Troncone, A.; Vena, M. 2021. "A simple approach for evaluating the bearing capacity of piles subjected to inclined loads." International Journal of Geomechanics (ASCE), 21(11): 04021224. DOI: 10.1061/(ASCE)GM.1943-5622.0002215.
Reviewer 2 Report
Please consider the following comments for your revision.
- "Steenbergen etal. [11] developed ~" should be corrected.
- "3.2. Internal vibration velocity response track and pied raft foundation" should be corrected.
- If the parameters in Table 9 was based on the references, please list them. The comparison of results between numerical analysis and model tests showed somewhat discrepancy in terms of velocity attenuation. Do you think the comparison can be enhanced by adjusting damping factors used?
- " The water content, degree of compaction and gradation size will have impact on vibration response characteristics of embankment and pile raft foundation. " What is the basis of this paragraph? I do understand the factors will impact on the behavior of pile raft foundation, but the variation of these parameters was not considered in this study in both testing and numerical analysis. So, it might be better to provide the justification for the sentences, otherwise remove the paragraph.
Round 2
Reviewer 1 Report
The replies provided by the authors are sufficient to address the issues the I raised during the previous round of review. Therefore, the manuscript could be accepted in its present form.